# Wild Boar Attacks on Hunting Dogs in Czechia: The Length of the Hunting Season Matters

**DOI:** 10.3390/ani15020130

**Published:** 2025-01-08

**Authors:** Jana Adámková, Karolína Lazárková, Jan Cukor, Hana Brinkeová, Jitka Bartošová, Luděk Bartoš, Kateřina Benediktová

**Affiliations:** 1Department of Game Management and Wildlife Biology, Faculty of Forestry and Wood Sciences, Czech University of Life Sciences Prague, 165 21 Praha, Czech Republic; lazarkova@fld.czu.cz (K.L.); brinkeova@fld.czu.cz (H.B.); bartos@vuzv.cz (L.B.); benediktovak@fld.czu.cz (K.B.); 2Department of Silviculture, Faculty of Forestry and Wood Sciences, Czech University of Life Sciences Prague, 165 21 Praha, Czech Republic; cukor@fld.czu.cz; 3Department of Game Management, Forestry and Game Management Research Institute, 252 02 Jíloviště, Czech Republic; 4Department of Ethology, Institute of Animal Science, 104 00 Praha, Czech Republic; bartosova.jitka@vuzv.cz

**Keywords:** *Canis familiaris*, *Sus scrofa*, injuries, animal aggression, wildlife management, seasonal evaluation, animal welfare

## Abstract

Driven hunts with hunting dogs are one of the tools for wild boar population reduction. However, they may represent an increased risk of injuries for hunting dogs. This study investigated whether hunting pressure, measured by the length of the hunting season, the frequency of hunts, and participant numbers, influenced the likelihood of wild boar attacks on dogs. The results revealed that the number of attacks reported in a hunting season only increased with longer hunting seasons. Other factors, such as shorter intervals between hunts, the number of driven hunts in the season, or the number of participants, did not enter the best statistical model in our study. Although most injuries were mild, severe and fatal cases were recorded, underscoring the need for improved strategies to reduce risks and enhance dog safety during hunts.

## 1. Introduction

In recent decades, the rising population of wild boars has drawn considerable attention [1,2,3]. This increase is connected to a rising rate of human–wildlife conflicts; significant damage to crops [4,5,6]; rooting [7,8,9]; or spreading diseases [10]—mainly African swine fever, currently [11,12,13,14,15,16]. These conflictual situations have resulted in efforts to reduce wild boar abundance as quickly as possible. A driven hunt is one of the most widely used and effective wild boar regulation methods [17,18,19,20]. Within a driven hunt, dogs play a crucial role in its success [21,22,23], but their activity can sometimes lead to injuries, as wild boars may attack them [21,24,25]. Wild boar attacks often target the thoracic and abdominal regions, requiring urgent surgical intervention [26]. Therefore, for dog owners, the factors leading to the incidence of hunting dogs’ injuries are a significant concern during driven hunts [21,24,27,28,29,30]. Previous studies indicated that larger dog breeds weighing over 20 kg are more likely to be injured, which suggests that wild boars perceive larger dogs as a more significant threat [21,24]. Conversely, smaller breeds, for example, dachshunds and terriers, tend to experience fewer injuries. However, injuries occurring in smaller breeds typically involve bold individuals [31]. These dogs are inclined to approach wild boars at close distances, and this daring behavior can prompt wild boars to feel threatened and attacked, even by small dogs that they would usually ignore.

For driven hunts, dogs are trained to flush, chase, track, or stop and hold wounded wild ungulates [23,25]. Wild boars may exhibit a physiological response similar to red deer when pursued by dogs, such as carbohydrate depletion, muscle tissue damage, and increased cortisol levels [32]. So, it is understandable that chased wild boars defend themselves [33]. Wild boars may attack hunting dogs, especially when they feel threatened, cornered, or injured [25].

Additionally, adverse weather conditions, such as heavy snow or intensive rainfall, may cause wild boars to remain hidden until the last moment, elevating the risk of direct contact with hunting dogs or hunters [23]. Defending wild boars use their tusks, the sharp lower tusks in particular, to inflict severe or even fatal injuries [25,34,35]. For many owners, hunting dogs are considered part of the family. Hence, incidents involving their harm represent not only a significant financial burden but also a source of psychological and emotional distress [36,37].

In general, several parameters that influence the effectiveness of driven hunts, including the number of hunters and their dogs present during the hunt; previous hunting success; or weather conditions such as temperature, wind, rain, or snow cover, have been studied by several authors, e.g., [23,38]. Among these factors, the local wild boar abundance is the main factor affecting hunting success, which is also influenced by the size, layout, and structure of the hunting grounds, which can multiply encounters with wild boars [17,39,40]. Furthermore, the number of hunters has been shown to correlate with hunting success positively [17,19]. Similarly, more hunting dogs present contribute to increased success rates [23]. However, the timing of driven hunts within the season has a contrasting effect, as hunting success tends to decline with the decreasing population density of wild boars later in the season [19]. Whereas the above-mentioned studies were focused on factors influencing hunting success and veterinary research documented injuries to hunting dogs inflicted by wild boars [21,24,35], no study has yet examined the factors affecting the risk of such attacks and injuries. Therefore, the primary aim of this study was to identify the factors associated with hunting pressure that contribute to attacks and injuries to dogs by wild boar during driven hunts in Czechia with a traditional approach to game management. We hypothesized more attacks on hunting dogs by wild boars with increasing hunting pressure on two different levels: (i) within a hunting season (i.e., a longer hunting season, more frequent driven hunts, and shorter intervals between hunts) and (ii) within a hunting event (i.e., higher number of wild boars harvested and higher number of participants, including hunters, beaters, and hunting dogs).

## 2. Materials and Methods

### 2.1. Study Area

Our research focused on the Benešov district, located in the southern part of the Central Bohemian region of the Czech Republic, covering an area of 1475 km^2^. The majority of this terrain consists of the Central Bohemian Highlands (mean elevation 362.9 m a.s.l.), characterized by a rugged, agricultural (61.3% of the district) and wooded (28.1%) landscape. The Benešov district includes 89 hunting grounds, spanning a total hunting area of 1306.16 km^2^, excluding non-hunting locations such as human settlements and roads.

### 2.2. Data Collecting Procedure

In April 2017, a retrospective questionnaire was distributed to all 89 hunting managers overseeing the hunting grounds in the Benešov district. We received responses from 50 hunting grounds (return rate 56.2%), from which ten were further excluded as incomplete. These forty valid hunting grounds (44.9% of all) covered a total area of 631.81 km^2^ (48.4% of the Benešov district hunting area, Figure 1) and ranged from 596 to 3055 hectares each. We collected data covering five consecutive hunting seasons from 2012 to 2016. In the Czech Republic, the official hunting year spans from 1 April to 31 March. In hunting grounds, however, the hunting season typically lasts two to four months into late autumn and winter, between October and January, with a driven hunt of a month. The driven hunts occur primarily during the day in agricultural and forested areas. In our study, ‘hunting season’ refers to the period between the first and the last months in which a driven hunt occurred (e.g., the first hunt in October and the last hunt in January result in a hunting season length of 4 months). Technical note: A hunting season spans two calendar years. In this study, the seasons were labeled by the starting year, e.g., the season running from October 2012 to January 2013 was labeled hunting season 2012.

### 2.3. Questionnaire Form

The questionnaire consisted of two sections. The first part gathered information on the hunting grounds (name, location, size, and forest area) and details of their driven hunts (numbers and distribution within the season and years, and number of participants including hunters, beaters, and dogs), while the second part collected data on wild boar (*Sus scrofa*) attacks on dogs of both sexes and all age categories, and the injuries sustained (numbers and severity of attacks). Three types of injuries were distinguished: mild injuries requiring no veterinary care, severe injuries that needed veterinary treatment, and fatal injuries resulting in the dog’s death. The complete questionnaire is shown in Appendix A. The questionnaire was set up to utilize mainly information routinely collected and maintained by the hunting grounds to avoid relying on respondents’ memory.

### 2.4. Ethics Statement

No personal data of the respondents were used in this study, ensuring privacy. Respondents provided detailed information about their hunting grounds, with their consent, and the data are also publicly available from the Forest Management Institute source “https://www.uhul.cz/portfolio/portalmyslivosti/ (accessed on 20 January 2024)”. All other data collected through the questionnaires were completely anonymous. By completing the online questionnaire available at survey.com, respondents consented to the use of their hunting ground data for this study.

### 2.5. Statistical Analysis

All data were analyzed using the SAS System (SAS, version 9.4). The individual countable metrics (see Table 1) were checked for possible multicollinearity (PROC CORR); significant correlations were found. We made a judgment of the extent of collinearity by checking related statistics, such as Tolerance Value, Variance Inflation Factor (VIF), Eigenvalue, and Condition Number, while using the TOL, VIF, and COLLIN options of the MODEL statement in the SAS REG procedure. Since the issues analyzed in this study represent more complex causality, we used the information-theoretic approach (IT-AIC) to estimate the factors’ effects on dependent variables [41].

Associations were subsequently sought between the dependent variable, i.e., a log-transformed number of wild boar attacks reported in a hunting ground within each hunting season, and the fixed factors (Table 1) using a multivariate General Linear Mixed Model (GLMM, PROC MIXED). The initial dataset included 200 rows (5 hunting seasons in 40 hunting grounds). Data from one hunting ground were excluded because it was only a 1-month hunting season, so the final sample size was 195. All analyses were performed using PROC MIXED, with the hunting ground ID as a random effect to account for repeated measures in the same hunting ground. We constructed 17 a priori hypotheses and added a null model. Where appropriate, we included interaction terms (Appendix A). For the dependent variable (i.e., number of attacks), we generated all GLMMs listed in Appendix A and converted the values of fit statistics.

We used expanded information criteria AIC, AICC, BIC, CAIC, and HQIC, available in SAS, to select a true model, as recommended by Christensen [42]. Then, we compared the candidate models by ranking them based on the information criteria (PROC RANK). The model with the lowest value (i.e., closest to zero) was considered to be the “best” model [41,42]. To see if the best model has merit, we compared our model to the null model for all dependent variables and all fitting criteria, showing delta (null—best model) and a relative information loss [exp ((null—best)/2)], an approach adapted from Burnham and Anderson [41].

The differences (Δ*_i_*) between the fit statistic values (the smallest values indicating the best-fitting model) were sorted according to the AIC, AICC, BIC, CAIC, and HQIC values. Akaike weight *w_i_* can be interpreted as the probability that M*_i_* is the best model (in the AIC sense, it minimizes the Kullback–Leibler discrepancy), given the data and the set of candidate models, e.g., [41]. For the five models with the lowest AIC values, we, therefore, calculated Δ AIC and Akaike weights *w_i_*, and for estimating the strength of evidence in favor of one model over the other, we divided their Akaike weights *w_min_*/*w_j_* (AIC Odds) [41]. This was also calculated for the additional fit statistics (i.e., AICC, BIC, CAIC, and HQIC).

Associations between the dependent variable and countable fixed effects are presented by fitting a random coefficient model using GLMM, as described by Tao et al. [43]. We calculated the predicted values of the dependent variable and plotted them against the fixed effects with predicted regression lines.

## 3. Results

During the analyzed seasons, 797 driven hunt events were organized (3.99 ± 1.43 in a hunting ground yearly; mean ± standard deviation). In those, 2891 wild boars were harvested (3.62 ± 3.05 per hunt).

In total, 589 attacks by wild boars on dogs were reported during driven hunts (2.60 ± 5.07 per hunting season in a hunting ground), of which 150 (25.5%) resulted in dog injuries. There were 0.77 ± 1.34 injuries per hunting season and hunting ground. The number of reported injuries and their severity are shown in Table 2. Injuries were predominately mild (73.8%) or severe (18.8%). Fatal injuries comprised 7.4% of cases. A dog participating in a driven hunt had a probability of 0.15 to be attacked and 0.039 to be injured (3634 dogs participated).

Table 3 and Appendix A show the five best candidate models that tested associations between the predictors and the number of attacked dogs, ranked by the five criteria for the best fit. All criteria ranked GLMM as the best. They did not differ when ranking the other candidate best models (Appendix A). Also, the differences (Δ) between the best and second-best models were the same for all the criteria (Δ for the second model, Δ AIC = 7.09, Δ AICC = 7.09, Δ BIC = 7.09; Δ CAIC = 7.09 Δ HQIC = 7.09). By comparing our best model to the null model, we have a convincing argument that the best model has merit with apparently negligible information loss estimated by all five fit criteria (Appendix A). Since fitting by all criteria was similar, we present further calculations for AIC only. The correct model’s probability was high (97%) compared to the second-best model (0.03%). The best-fitting GLMM was thus 34.59 times (odds) more likely to be the correct model than the second-best model.

According to the best model, the number of wild boar attacks on dogs was affected only by the length of the hunting season (see Figure 2). Appendix A shows estimates, standard error, and 95% confidence interval for the best-fitting GLMM model for this dependent variable.

## 4. Discussion

As predicted (i), the number of wild boar attacks on hunting dogs reported during the hunting season increased with the length of the hunting season. The longer the hunting season, the more attacks. A longer hunting season simply increases the likelihood of interactions between dogs and wild boars, increasing the risk of injuries. This aligns with our assumption that such attacks are inevitable, given sufficient time and repeated opportunities.

Interestingly, any other of the tested factors hypothesized to heighten hunting pressure, including intervals between driven hunts; hunting frequency within the same area [17,18]; or the number of wild boars harvested and the number of participants, i.e., hunters, beaters, and dogs [19,23], were not included in the model best-describing effects associated with the number of attacks on hunting dogs. Nevertheless, these variables were often low or moderately correlated with the length of the hunting season, e.g., the number of driven hunts in the season or intervals between hunts. Thus, the conclusion that the length of the season was the best representative of a mutually correlated bunch of variables can be made, rather than the variables that did not assert themselves are not of influence.

While cumulative stress and hunting pressure have been highlighted in previous studies, these factors alone may not dictate attack patterns. For instance, Kokkinos et al. [35] observed more frequent wild boar attacks on dogs in December, a period consistent with our findings of increased attack likelihood late in the season. Notably, hunting success tends to peak early in the season, with the number of dogs and beaters showing little effect on wild boar harvest rates [19]. This late-season surge in attacks may reflect heightened defensive responses from wild boars following repeated exposure to hunting events. However, our study was limited in such a detailed analysis because the number of attacks in particular months was unavailable in our study.

Repeated driven hunts in the same or neighboring hunting grounds expose wild boars to sustained pressure [17,18,44], potentially encouraging them to develop avoidance strategies, such as hiding or fleeing [20,45,46]. More experienced individuals are generally more likely to escape than confront dogs [44]. However, wild boars may resort to defensive behaviors when escape is impossible—due to being cornered, injured, or exhausted. These situations may potentially lead to attacks on hunting dogs.

Alternatively, wild boars may choose not to flee because they feel confident. In our study, approximately 4.56 ± 2.66 (mean ± SE, Table 1) dogs were used in driven hunts, indicating a low number of dogs and high homogeneity of the data, which may explain why the effect of the number of dogs did not assert itself in the best model. However, it is also possible that the behavior of wild boar, especially their tendency to flee when approached by a dog, is less influenced by the number of hunting dogs and more dependent on the quality of the dogs’ training [18]. Conversely, wild boars may also confront dogs, possibly due to confidence gained from previous positive encounters with small dogs, as noted by Thurfjell et al. [45], or with fewer dogs, as observed in our study. Wild boars may be able to confront dogs not only because of the confidence gained from previous encounters but also because of their superior size, weight, and sharper tusks, especially in males [21,34,35]. Population structure may further explain this behavior. As the hunting season progresses in the same district, the population of piglets and yearling boars decreases due to selective hunting, so larger and more experienced animals prevail. Moreover, piglets and yearlings gain weight and skills compared to the beginning of the driven hunt season. Therefore, the observed increase in attacks in longer seasons could be attributed to this shift in population dynamics.

Other factors not investigated in the present study might also influence the attacks. Climatic conditions, particularly snow cover, can significantly influence wild boar movements [47]. As the hunting season progresses, the energy-saving benefits of staying hidden during harsh weather may outweigh the risks of being detected. Wild boars often seek shelter and protection in adverse weather, making their detection harder for dogs and hunters [17]. Heavy snow cover can also be a physical barrier, limiting wild boar movement and increasing the risk of direct encounters with hunting dogs. Wild boars remain hidden during rainy conditions, while hunters may be less proactive, affecting shooting accuracy [3,40]. This can increase the likelihood of severe injuries rather than successful harvests. Conversely, frosty, sunny weather may improve hunter morale and accuracy, although wild boars might flush more easily.

In addition to environmental factors, the dog’s physical condition plays a crucial role in the likelihood of injuries. As the hunting season progresses, cumulative fatigue and slower recovery between hunts may reduce a dog’s agility and reaction times during confrontations with wild boars.

A study based on data collected via a retrospective questionnaire has limitations in terms of reliability. Thus, our questionnaire was carefully set up to minimize this flaw using principally data from hunting grounds evidence. Hunting managers collect and maintain records as the Hunting Law requires (see questions 3–10 in Appendix A). Hunting managers also typically issue certificates for dog owners to insurance companies if a dog sustains injuries during hunting (questions 11–14 in Appendix A). Despite the inherent limitations of this kind of study, our findings provide reliable insights into the factors influencing dog injuries during driven hunts. The outcomes of the present study will serve as a foundation for a detailed follow-up study involving data collection on particular driven hunt events.

## 5. Conclusions

This study highlights the growing risk of injuries to hunting dogs during driven hunts, with the length of the hunting season emerging as a key factor. In our research, longer hunting seasons were associated with more attacks. The injuries were mostly mild, but a quarter were severe and fatal, emphasizing the need for improved hunting strategies. A deeper investigation of the relationship between external factors and dog injuries is needed to provide insight into the behavioral dynamics of wild boar–dog interactions. Our findings thus do not suggest effective measures to prevent wild boar attacks on hunting dogs yet.

## Figures and Tables

**Figure 1 animals-15-00130-f001:**
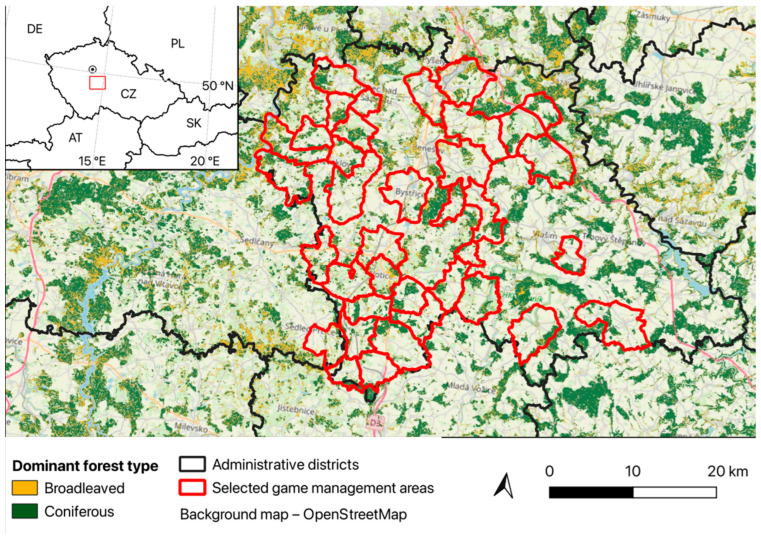
The study area map shows 40 hunting grounds (game management areas) with valid responses and the distribution of the forested regions and agricultural fields. EN: © EuroGeographics for the administrative boundaries. OpenStreetMap. Source service: © CENIA, česká informační agentura životního prostředí, Source data: © Agentura ochrany přírody a krajiny, Available online: Národní geoportál INSPIRE http://geoportal.gov.cz (accessed on 27 December 2024).

**Figure 2 animals-15-00130-f002:**
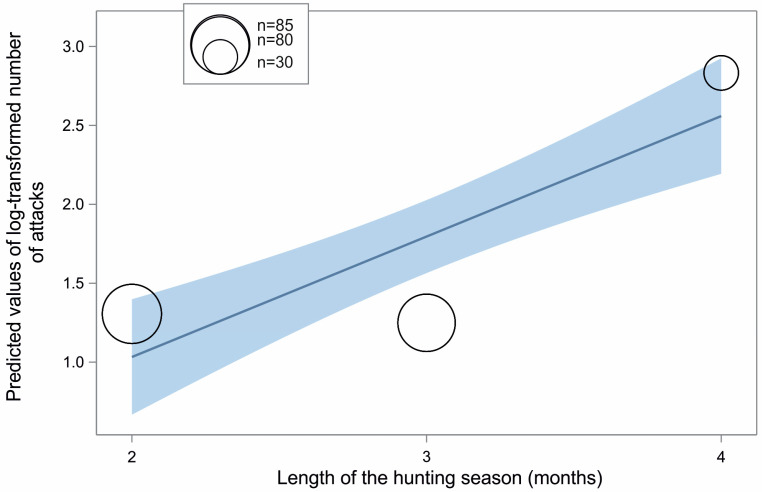
Predicted values of the number of wild boar attacks reported during a hunting season on dogs (log-transformed) with 95% confidence intervals according to the hunting season length (*x*-axis). The bubble size refers to the number of hunting seasons the data were obtained from (n = 195).

**Table 1 animals-15-00130-t001:** Summary of potential fixed factors impacting the number of dog attacks by wild boar, including the mean and standard deviation. The ‘n’ column specifies if one value from a hunting ground was obtained for overall hunting seasons (n = 40) or for each season (n = 200).

Countable Variables			
Variable	Mean	Std Deviation	n
Area of hunting ground (ha)	1579.53	599.70	40
Forest area of hunting ground (ha)	390.65	178.70	40
Number of participants (beaters and hunters) per driven hunt	23.8	10.69	40
Number of beaters per driven hunt	7.6	4.53	40
Number of hunters per driven hunt	16.2	8.65	40
Number of dogs per driven hunt	4.56	2.66	40
Number of driven hunts per hunting season	3.99	1.43	200
Length of the hunting season in a hunting ground (months)	2.68	0.76	40
Interval between driven hunts in a hunting ground within the hunting season (days)	17.85	4.83	40
Number of wild boars harvested in a hunting ground per hunting season (pcs)	14.46	13.10	200
Categorial variables			
Hunting ground ID name	40 hunting grounds		
Hunting season	1–5 (2012–2016)		
Month	October, November, December, January		

**Table 2 animals-15-00130-t002:** Summary of the number and types of injuries recorded over five hunting seasons.

Hunting Season	October			November			December			January		
	Mild	Severe	Fatal	Mild	Severe	Fatal	Mild	Severe	Fatal	Mild	Severe	Fatal
2012–2013	0	0	0	6	1	2	14	4	3	0	0	0
2013–2014	1	0	0	7	1	3	9	8	3	0	0	0
2014–2015	0	0	0	9	1	0	15	0	0	3	3	0
2015–2016	0	0	0	11	1	0	11	0	0	2	0	0
2016–2017	0	1	0	2	1	0	20	6	0	0	2	0
total	1	1	0	35	5	5	69	18	6	5	5	0

**Table 3 animals-15-00130-t003:** Five best-fitting models were sorted according to fit AIC (the smaller, the better), AIC difference (Δ*_i_*), AIC weight (*w_i_*), and AIC Odds for the dependent variable log-transformed number of attacks. The parentheses indicate factors nested in hunting season.

Model	AIC	AIC Δ*_i_*	AIC *w_i_*	AIC ODDs
The length of the season	416.81	0.00	0.97	1.00
The length of season (hunting season)	423.90	7.09	0.03	34.59
The length of season (hunting season), the interval between driven hunts	429.73	12.92	0.00	638.90
The length of season (hunting season), the length of season interval between driven hunts	429.73	12.92	0.00	638.90
Number of dogs (hunting season)	432.81	15.99	0.00	2972.87

## Data Availability

The raw data supporting the conclusions of this article will be made available by the authors on request.

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
