# Peer review of "Wild Boar Attacks on Hunting Dogs in Czechia: The Length of the Hunting Season Matters"

_animals, 2025, doi:10.3390/ani15020130_

Round 1

Reviewer 1 Report

Comments and Suggestions for Authors

Thank you for the opportunity to review the manuscript entitled “Wild boar attacks on hunting dogs: Why the length of the hunting season matters”. I found the manuscript very interesting and worthy of being published. I have a few comments but in my opinion, the manuscript needs only minor revision. I think the Methods are a little bit messy and used variables could be better explained. All the comments are listed below. I wish the Authors all the best.

Comments:

-P.1 l.14-16: It’s repetition in two first sentences. I suggest joining them, for example: As wild boar populations grow, so does the risk of injury to hunting dogs, which play a crucial role in managing wild boar numbers within driven hunts. The same in the Abstract (p. 1, l. 22-28).

In my opinion, the Simple Summary and the Abstract are almost the same. I suggest changing them, or one of them. According to the Instructions for Authors, Simple Summary should be written for a lay audience and show how they will be valuable to society. I think it is well written, but I would skipped the first and the third sentence (“While veterinary reports have detailed these…”).

p.1 l. 43 I wouldn’t say “characterized” but rather “is connected”.

p. 2 l. 46-47 Why is the sentence bolded?

p. 2 l. 76 what does “layout” mean here?

p. 2 l. 90 What does “extended hunting season” mean? How do you understand more driven hunts? More in total? Increasing number of hunts per day? Increasing risk for wild boar? Shorter intervals between hunts – do you mean days? Hours within one event? It is used in the context of Central Europe, so it must be taken from more than one hunting ground. Thus, it’s hard for me to imagine how the hunting pressure was measured.

p. 3 l. 97 So it’s not the whole Central Europe. In that case, I would not write “Central Europe” in the aim of the study.

p. 3 l. 110 Is it possible to give the dates for the driven hunt month?

You can skip the brackets in the last sentence.

Methods – I didn’t find the description of the variables used in the statistical analyses. As I mentioned before, what do you mean by hunting season? The whole season or just driven hunt month/months. Length of the hunting season in a hunting ground (months) – how long it should be. The other variables are clear when I look at the table.

What does the n mean in Table 1? Total number of dogs/ participants etc. or number of cases (filled questionnaires)? I already understood that only questionnaires with all the answers were used, so I do not know what this value means.

p. 4 l. 139-146. I do not understand: first the models were performed and then the hypotheses?

I don’t understand Supplementary Table S2. What is the key to this order? Again, I suggest adding the characteristics of variables and interactions in the main text.

p. 4 l. 147 Do you mean Table S3? In Table 3 the results for all models should be given.

Why the Authors did not create one general model, instead of 18 “small” ones?

Figure 1 is too hard to understand for me. I don’t see the differences between circles, I don’t understand the legend. The hunting season wasn’t important in the model, so it can be skipped in the graph.

p. 7 l. 252-265 It was not tested in the research.

I don’t know if it’s possible, but I would like to see if any of the variables affect the severity of injuries.

Author Response

Response to Reviewer 1 Comments

1. Summary

Thank you for the opportunity to review the manuscript entitled “Wild boar attacks on hunting dogs: Why the length of the hunting season matters”. I found the manuscript very interesting and worthy of being published. I have a few comments but in my opinion, the manuscript needs only minor revision. I think the Methods are a little bit messy and used variables could be better explained. All the comments are listed below. I wish the Authors all the best.

Response: Thank you for taking the time to review our manuscript and for your positive evaluation. We greatly appreciate your helpful comments and suggestions. We have carefully addressed each point and revised the manuscript accordingly. Detailed responses are provided below (in blue following “Response”), and all changes are tracked in the resubmitted manuscript.

We recalculated the statistical model in response to the Reviewers‘ comments. While the results remain consistent, we have improved the clarity of their presentation both in the text and through enhanced graphical representations.

2. General Evaluation

Response: We rearranged and completed all the parts evaluated as “Can be improved” (i.e., Methods descriptions, Results) as suggested by the reviewer in detailed comments.

3. Point-by-point response to Comments and Suggestions for Authors

Comments 1:

P.1 l.14-16: It’s repetition in two first sentences. I suggest joining them, for example: As wild boar populations grow, so does the risk of injury to hunting dogs, which play a crucial role in managing wild boar numbers within driven hunts. The same in the Abstract (p. 1, l. 22-28).

Response 1: Thank you for pointing this out. Reworded.

Comments 2: p.1 l. 43 I wouldn’t say “characterized” but rather “is connected”.

Response 2: Thank you, completed.

Comments 3: p. 2 l. 46-47 Why is the sentence bolded?

Response 3: Because of incorrect text format setting; corrected. Thanks for being so attentive.

Comments 4: p. 2 l. 76 what does “layout” mean here?

Response 4: Arrangement, i.e. how the hunting grounds are organized or positioned in a space. We preferred to keep the term “layout” because it was stated in cited papers.

Comments 5: p. 2 l. 90 What does “extended hunting season” mean? How do you understand more driven hunts? More in total? Increasing number of hunts per day? Increasing risk for wild boar? Shorter intervals between hunts – do you mean days? Hours within one event? It is used in the context of Central Europe, so it must be taken from more than one hunting ground. Thus, it’s hard for me to imagine how the hunting pressure was measured.

Response 5: We agree it was confusing and reformulated. We had data from 40 hunting grounds.

Comments 6: p. 3 l. 97 So it’s not the whole Central Europe. In that case, I would not write “Central Europe” in the aim of the study.

Response 6: Agree; Central Europe changed to Czechia.

Comments 7: p. 3 l. 110 Is it possible to give the dates for the driven hunt month?

            You can skip the brackets in the last sentence.

Response 7: Thanks for this point. In the manuscript, we specified the driven hunt months.

           Brackets were deleted.

Comments 8: Methods – I didn’t find the description of the variables used in the statistical analyses. As I mentioned before, what do you mean by hunting season? The whole season or just driven hunt month/months. Length of the hunting season in a hunting ground (months) – how long it should be. The other variables are clear when I look at the table.

Response 8: Thank you for your comment. The definition of “hunting season” was added to the manuscript. In our study, "hunting season" refers to the whole season, i.e., the period between the first and the last months in which a driven hunt occurred (e.g., first hunt in October, last hunt in January, length of the season = 4 months).

Comments 9: What does the n mean in Table 1? Total number of dogs/ participants etc. or number of cases (filled questionnaires)? I already understood that only questionnaires with all the answers were used, so I do not know what this value means.

Response 9: The ‘n’ column in Table 1 specifies if one value was obtained for a hunting ground (n = 40) or if there were 5 values for each of the 40 hunting grounds (n = 200). We agree that it was confusing for the readers. We explained it in the ‘Questionnaire form’ section and added information to the Table 1 capture.

Comments 10: p. 4 l. 139-146. I do not understand: first the models were performed and then the hypotheses?

Response 10: Thank you for your comment. We appreciate your feedback, but we may not have fully understood the specific concern raised in this instance. To clarify, two hypotheses were formulated at the end of the Introduction. Additionally, the text on lines 139–144 outlines a fundamental check typically conducted when working with multiple countable variables. This step helps to prevent potential issues arising from multicollinearity or mutual dependence between variables within a multivariable model.

The confusion was most likely caused by the misleading word “then”; it was deleted. In the IT-AIC approach, “a priori hypotheses” are constructed as relevant combinations of factors with expected (potential) impact on the dependent variable.

Comments 11: I don’t understand Supplementary Table S2. What is the key to this order? Again, I suggest adding the characteristics of variables and interactions in the main text.

Response 11: Thank you very much for your comment. Variable characteristics are shown in Table 1. According to our belief, a complete set of a priori hypotheses (within the IT-AIC approach) fits better as Supplementary material. There are 5 best models listed in Table 3. Still, Table S2 can be moved to the main body of the manuscript if found more appropriate within the editorial process.

We utilized an established statistical procedure, "Model Selection Using Akaike's Information Criterion," for our data analysis. While we respect the reviewer's perspective, it's important to note that providing a succinct explanation of this concept is challenging, given the extensive literature surrounding it (bare references can be found in the manuscript).

In essence, the Akaike Information Criterion (AIC), which we have extended to incorporate four additional criteria (detailed in the Methods section), is a statistical tool for model selection. It effectively balances the model's goodness of fit with its complexity, allowing us to identify the best-fitting model among a group (the "small ones") by, for example, penalizing for the number of parameters. This approach helps prevent overfitting, ensuring our conclusions are robust. We opted for this methodology because it resolves some limitations of traditional null hypothesis testing, which can sometimes lead to misleading p-values. Such statistical strategies have recently been strongly recommended for our data type.

Comments 12: Why the Authors did not create one general model, instead of 18 “small” ones?

Response 12: Thank you for your comment. We appreciate your feedback and would like to kindly direct your attention to our earlier response, where we have addressed this point in detail.

Comments 13: p. 4 l. 147 Do you mean Table S3? In Table 3 the results for all models should be given.

Response 13: Thanks for this point. Table S2 is correct (on p. 4, l. 147 of the original manuscript); the word “listed” was added to clarify the sentence.

According to the established approach, the fitting of 5 best models and a null model are shown (Table 3 and Table S3). The estimated parameters of the factor involved in the best model are in Table S5. In the main text, Figure 2 represents the main result.

The IT-AIC approach cannot be combined with classical statistics. This is why no F or p values are associated with factors included in the best model.

Comments 14: Figure 1 is too hard to understand for me. I don’t see the differences between circles, I don’t understand the legend. The hunting season wasn’t important in the model, so it can be skipped in the graph.

Response 14: Thank you for this point. Based on all the reviewers’ comments, we found our mistake in preparing a priori hypotheses; the model, including only the effect of the season’s length (not nested in the season ID), was missing. We did the analyses again and it was as expected. The best model was that containing the length of the season only. Thus, the hunting season disappeared from Figure 1 (now Figure 2), which is now simpler and better understandable.

The size of the circles in the graph represents the sample size, i.e., the number of hunting seasons the data comes from (195 altogether, 5 seasons in 39 hunting grounds; one hunting ground was excluded from analyses because it was only ground in that hunting season lasted only on month).

Comments 15: p. 7 l. 252-265 It was not tested in the research.

Response 15: Thanks for the comment. You are right; it was not tested in our study. However, we considered it important to notice further potential, influential factors not investigated in our study. We indicated it in the manuscript.

Comments 16: I don’t know if it’s possible, but I would like to see if any of the variables affect the severity of injuries.

Response 16: Thank you for a good point. Unfortunately, the data distribution did not allow a reliable analysis. Based on the results, we plan a deeper study involving data collection about individual-driven hunts. That is why we skipped this in the manuscript.

4.     Response to Comments on the Quality of English Language

Response: The quality of English does not limit understanding of the research by Reviewer 1. However, addressing numerous comments from Reviewer 1 helped improve the comprehensiveness and English.    

Reviewer 2 Report

Comments and Suggestions for Authors

The title is not appropriate because the research results did not provide an answer to the question "Why the length of the hunting season matters" but only guesses and assumptions were described.

The quality of the manuscript would be much better if you provided a little more information about the obtained results in the results and discussion. Are there data on the total number of injuries during the hunting season or are there data that the number was higher at the very end of the season?

In the discussion, it is said that the number of attacks on dogs was higher at the end of the hunting season, although there is no data in which month exactly the attacks took place, it is only guessed, and not based on the interpretation of the obtained results. Although in the questionnaire (the second part of collected data: numbers and severity of attacks) there is a question regarding the severity of the injury and the month when it occurred, the authors state that they do not have data about the month when the attacks occurred (line 223)? Please clarify that.

Reading the manuscript, the reader gets the impression that the obtained results and the discussion are not connected, that there are no clear and precise conclusions, but everything is just an assumption.

The lack of research is that there is no data on any specific event of the attack itself, or under what circumstances it happened.

In the discussion, statements from the introduction are repeated (for example: lines 62-64 and 231-234).

Author Response

Response to Reviewer 2 Comments

1. Summary

Thank you for taking the time to review our manuscript and for your positive evaluation. We greatly appreciate your helpful comments and suggestions. We have carefully addressed each point and revised the manuscript accordingly. Detailed responses are provided below (in blue under “Response”), and all changes are tracked in the resubmitted manuscript.

We recalculated the statistical model in response to the Reviewers‘ comments. While the results remain consistent, we have improved the clarity of their presentation both in the text and through enhanced graphical representations.

2. General Evaluation

Response: According to your comments, we rearranged and completed all the parts evaluated as “Can be improved” (i.e., Research design, Methods descriptions) and "Must be improved" (i.e., Results, Conclusions).

3. Point-by-point response to Comments and Suggestions for Authors

Comments 1: The title is not appropriate because the research results did not provide an answer to the question "Why the length of the hunting season matters" but only guesses and assumptions were described.

Response 1: Thank you for this comment. We agree with your suggestion and modified the title.

Comments 2: The quality of the manuscript would be much better if you provided a little more information about the obtained results in the results and discussion. Are there data on the total number of injuries during the hunting season or are there data that the number was higher at the very end of the season?

Response 2: We agree and improved the Methods and Discussion accordingly. The data was collected as a number of attacks per season in a hunting ground. Thus, any conclusion on the dynamics of attacks within the season cannot be made. Detailed table with numbers of injuries was added (now Table 2).

Comments 3: In the discussion, it is said that the number of attacks on dogs was higher at the end of the hunting season, although there is no data in which month exactly the attacks took place, it is only guessed, and not based on the interpretation of the obtained results. Although in the questionnaire (the second part of collected data: numbers and severity of attacks) there is a question regarding the severity of the injury and the month when it occurred, the authors state that they do not have data about the month when the attacks occurred (line 223)? Please clarify that.

Response 3: We appreciate this comment. Sorry for the confusing expression; it was changed, and the Discussion was carefully checked for improper interpretations. Ad the data structure: We have the monthly data only on injuries. The attacks were collected as a count for a hunting season.

Comments 4: Reading the manuscript, the reader gets the impression that the obtained results and the discussion are not connected, that there are no clear and precise conclusions, but everything is just an assumption.

Response 4: Thanks for this comment. We adjusted the text to limit such an impression.

Comments 5: The lack of research is that there is no data on any specific event of the attack itself, or under what circumstances it happened.

Response 5: This is a very good point. Thanks. The present study aimed to identify factors associated with hunting pressure and wild boar attacks. It serves as a base for the deeper follow-up study involving data collection about particular driven hunts.

Comments 6: In the discussion, statements from the introduction are repeated (for example: lines 62-64 and 231-234).

Response 6: We appreciate this attentive comment. Repeated parts were deleted.

4.       Response to Comments on the Quality of English Language

Response: The quality of English does not limit understanding of the research by Reviewer 2. However, addressing numerous comments from Reviewer 2 helped improve the comprehensiveness and English.

Reviewer 3 Report

Comments and Suggestions for Authors

Wild boar attacks on hunting dogs: Why the length of the hunting season matters

The manuscript through a retrospective questionnaire survey in 2017 from 40 hunting grounds in the Benešov district, Central Bohemian region of the Czech Republic obtaining 150 incidences of wild-boar injuries on hunting dogs took place in 767 driven hunts between 2012 and 2016, predicted the length of the hunting season the key factors that contribute to these attacks.

While secondary data have its own inherent limitations, the retrospective survey adds more negatives to it. Over and above retrospective survey from past years adds much more uncertainty. Given this background, how far authors are guaranteed that hunting managers would have recalled situations / incidents took place since the past five years and answered accurately for the questions no starting from 3 to 14, in case they do not maintain all these details?  Do they maintain all these in a database systematically and if so, please mention it in the methodology.

The wild-boar attack on hunting dog or hunters, need not to be decided only by the variables such as period of hunting season, the number of wild boars harvested, the duration of hunting season, the frequency of hunts, and the number of participants and dogs involved, as considered by the authors in the study, Other factors like the hunting location (wooded forest, agricultural field, open areas), hunting time including moon-light, hunter’s age, and their experience, the hunting dog size or breed, its experience, its age-sex classes including the presence or absence of pregnancy of female dogs etc. Similarly, if hunters go far hunting both male and female wild pigs, the sex and age of the pig also matters. While much of these are unable to covered by the study, as it relied on retrospective that too for a long period, how the present study is going to be comprehensive than those carried out earlier?         

Line No. 231: The study cites earlier study that stated a gunshot-injured wild boar may experience heightened aggression towards dogs, driven by pain and stress associated with the injury [32].

Further following I make following specific comments and suggestions and incorporating them would improve the clarity of the manuscript.  

The title does not convey the location / area where the study was carried out.

Both Summary and Abstract are almost the same. While the summary supposed to be simple lay man standard writing, abstract must be highly technical with observed major findings, possible reasons for the same and measures to mitigate the same also be suggested.

Further, though the abstract says that this study examines key variables associated with dog injuries, including the timing and duration of the hunting season, the number of wild boars harvested, hunt frequency, and the number of participants and dogs involved. I don’t see much findings / results related the most variables mentioned above, which is a must for the abstract.

Introduction: It must review all the earlier studies on this aspect and justify on what ground this study is necessary.

Line No. 81-82: Despite these well-studied relationships, research has yet to  address how these factors impact the risk of injuries to hunting dogs during driven hunts.  This justification is not sufficient to warrant a new study.  

Study area: Is it possible to produce a map of the study area showing various land use and landcover details to understand of the hunting grounds?

Method: The study deals with wild boar throughout the text. The term wild boar refers to male wild pig. While both male and female pigs are in conflict with human everywhere, why this paper deals only with male wild big. Is it only the male pig alone involves in conflict with people in the study area and hence the hunters control males through hunting or the males alone permitted for hunting or it is by mistake wild boar instead of wild pig. This needs to be clearly described in method section.

Further, method section needs more details as to where hunting is takes place in terms of LULC (Is it only in agricultural areas or forested areas too), when the hinting takes place, whether during day or night.

There should also be details about the dog breeds used, their age, sex and reproductive state etc.

Results: Ok.

Discussion: Discussion should also highlight caveats in the present study, more specifically what all the ways the retrospective survey ended up in drawbacks in its findings and how one can over these drawbacks.   

Conclusions:

Line No. 266-268: The study highlights the growing risk of injuries to hunting dogs during driven hunts due to rising wild boar populations, with the length of the hunting season emerging as a key factor.

For the above statement, in conclusion section needs data support, which means this study should show that empirical data increasing injuries to hunting dogs with increasing wild boar / pig populations. When this study does not have data on wild boar or pig population and injury to hunting dog population over the years, the sentence needs to rephrased accordingly.

Apart from stating ‘A deeper investigation of the relationship between external factors and dog injuries is needed to provide insight into the behavioral dynamics of wild  boar-dog interactions’ what are measures the current study suggest to reduce wild boar or pig injury on hunting dog, which is essential.      

Author Response

Response to Reviewer 3 Comments

1. Summary

The manuscript through a retrospective questionnaire survey in 2017 from 40 hunting grounds in the Benešov district, Central Bohemian region of the Czech Republic obtaining 150 incidences of wild-boar injuries on hunting dogs took place in 767 driven hunts between 2012 and 2016, predicted the length of the hunting season the key factors that contribute to these attacks.

Response: Thank you for taking the time to review our manuscript and for your positive evaluation. We greatly appreciate your helpful comments and suggestions. We have carefully addressed each point and revised the manuscript accordingly. Detailed responses are provided below (in blue following “Response”), and all changes are tracked in the resubmitted manuscript.

We recalculated the statistical model in response to the Reviewers‘ comments. While the results remain consistent, we have improved the clarity of their presentation both in the text and through enhanced graphical representations.

2. General Evaluation

Response: According to your comments, we rearranged and completed all the parts evaluated as “Can be improved” (i.e., Introduction, Research design, Methods descriptions) and "Must be improved" (i.e., Conclusions).

3. Point-by-point response to Comments and Suggestions for Authors

Comments 1: While secondary data have its own inherent limitations, the retrospective survey adds more negatives to it. Over and above retrospective survey from past years adds much more uncertainty. Given this background, how far authors are guaranteed that hunting managers would have recalled situations / incidents took place since the past five years and answered accurately for the questions no starting from 3 to 14, in case they do not maintain all these details?  Do they maintain all these in a database systematically and if so, please mention it in the methodology.

Response 1: Thank you for your insightful comment. We acknowledge the inherent limitations of secondary data and have added a sentence with a statement addressing this issue in the revised manuscript. For questions 3–10, hunting managers maintain records as the Hunting Law requires. Regarding the second part of the questionnaire, which focuses on "Wild Boar Attacks" (questions 11–14), hunting managers typically issue certificates for dog owners to provide to insurance companies if a dog sustains injuries during hunting. Therefore, we assumed the managers had an adequate overview of such incidents when completing the questionnaire.

It is important to emphasize that this study was intended as a preliminary investigation to assess the feasibility of conducting more detailed and methodologically refined research in the future.

Comments 2: The wild-boar attack on hunting dog or hunters, need not to be decided only by the variables such as period of hunting season, the number of wild boars harvested, the duration of hunting season, the frequency of hunts, and the number of participants and dogs involved, as considered by the authors in the study, Other factors like the hunting location (wooded forest, agricultural field, open areas), hunting time including moon-light, hunter’s age, and their experience, the hunting dog size or breed, its experience, its age-sex classes including the presence or absence of pregnancy of female dogs etc. Similarly, if hunters go far hunting both male and female wild pigs, the sex and age of the pig also matters. While much of these are unable to covered by the study, as it relied on retrospective that too for a long period, how the present study is going to be comprehensive than those carried out earlier? 

Response 2: Thank you for your valuable comment. We fully agree that the variables considered in our study are not the only factors that may influence wild boar attacks. As noted in our earlier response, this preliminary study relied on a retrospective questionnaire and secondary data, which limited our ability to incorporate additional factors, such as hunter experience, dog characteristics (e.g., breed, size, sex, age, pregnancy), and wild boar demographics (e.g., sex and age).

We acknowledge that these variables could provide further insights into the dynamics of wild boar aggression and should be considered in future research employing more targeted and systematic data collection methods.

Comments 3: Line No. 231: The study cites earlier study that stated a gunshot-injured wild boar may experience heightened aggression towards dogs, driven by pain and stress associated with the injury [32].

Response 3: Thank you for bringing this to our attention. We appreciate the reviewer's suggestion and acknowledge that this study may not directly relate to prior research in the field.

Comments 4: The title does not convey the location / area where the study was carried out.

Response 4: Thank you. We added Czechia to the title.

Comments 5: Both Summary and Abstract are almost the same. While the summary supposed to be simple lay man standard writing, abstract must be highly technical with observed major findings, possible reasons for the same and measures to mitigate the same also be suggested.

Response 5: Thank you for your comment; we rewrote both the Summary and Abstract.

Comments 6: Further, though the abstract says that this study examines key variables associated with dog injuries, including the timing and duration of the hunting season, the number of wild boars harvested, hunt frequency, and the number of participants and dogs involved. I don’t see much findings / results related the most variables mentioned above, which is a must for the abstract.

Response 6: Thank you for your comment; completed.

Comments 7: Introduction: It must review all the earlier studies on this aspect and justify on what ground this study is necessary.

Response 7: Thank you for your comment. Completed.

Comments 8: Line No. 81-82: Despite these well-studied relationships, research has yet to address how these factors impact the risk of injuries to hunting dogs during driven hunts.  This justification is not sufficient to warrant a new study.

Response 8: Thank you for your comment. We acknowledge that our justification was insufficient. Therefore, we have revised the text to explain the necessity of our study better. To our knowledge, existing research has primarily focused on hunting effectiveness or documented the occurrence and severity of injuries to hunting dogs. However, no studies have specifically investigated the factors that increase the likelihood of wild boar attacks on dogs.

Comments 9: Study area: Is it possible to produce a map of the study area showing various land use and landcover details to understand of the hunting grounds?

Response 9: A map of the study area was added.

Comments 10: Method: The study deals with wild boar throughout the text. The term wild boar refers to male wild pig. While both male and female pigs are in conflict with human everywhere, why this paper deals only with male wild big. Is it only the male pig alone involves in conflict with people in the study area and hence the hunters control males through hunting or the males alone permitted for hunting or it is by mistake wild boar instead of wild pig. This needs to be clearly described in method section.

Response 10: Thank you for your comment. It was added to the Methods that by wild boar we mean males and females of all ages.

Comments 11: Further, method section needs more details as to where hunting is takes place in terms of LULC (Is it only in agricultural areas or forested areas too), when the hinting takes place, whether during day or night.

Response 11: Thank you. We added a description and a map to the methodology. Detailed information was collected, including the proportions of arable land and forest within the hunting areas (see Table 1 and Supplementary Table S2). Nevertheless, GLMMs incorporating these factors did not rank among the top five models. Notably, the forested area was identified as the ninth-best model based on all fitting criteria.

Comments 12: There should also be details about the dog breeds used, their age, sex and reproductive state etc.

Response 12: Thank you for your comment. Unfortunately, it was impossible to obtain such detailed information via the retrospective questionnaire survey and secondary data. We agree with you that this information is another key factor that can influence the aggressiveness of wild boars. We plan to collect this kind of data in the follow-up study.

Comments 13: Discussion should also highlight caveats in the present study, more specifically what all the ways the retrospective survey ended up in drawbacks in its findings and how one can over these drawbacks.   

Response 13: Thank you for your attentive comment. Completed in the manuscript.

Comments 14: Line No. 266-268: The study highlights the growing risk of injuries to hunting dogs during driven hunts due to rising wild boar populations, with the length of the hunting season emerging as a key factor.

For the above statement, in conclusion section needs data support, which means this study should show that empirical data increasing injuries to hunting dogs with increasing wild boar / pig populations. When this study does not have data on wild boar or pig population and injury to hunting dog population over the years, the sentence needs to rephrased accordingly.

Response 14: Thank you for a good point. We apologize for being inaccurate in expression. Completed; population rise removed from the manuscript.

Comments 15: Apart from stating ‘A deeper investigation of the relationship between external factors and dog injuries is needed to provide insight into the behavioral dynamics of wild boar-dog interactions’ what are measures the current study suggest to reduce wild boar or pig injury on hunting dog, which is essential.   

Response 15: Very apt point. Actually, we can say which factors do not matter as only trivial relationships “the longer period, the more attacks occur” asserted itself in the best model. Thus, the fewer opportunities, the fewer attacks/injuries.

4.     Response to Comments on the Quality of English Language

Response: The quality of English does not limit understanding of the research by Reviewer 3. However, addressing numerous comments from Reviewer 3 helped improve the comprehensiveness and English.

Round 2

Reviewer 2 Report

Comments and Suggestions for Authors

Thank you for accepting my suggestions.